# Lactose Content and Selected Quality Parameters of Sheep Milk Fermented Beverages during Storage

**DOI:** 10.3390/ani12223105

**Published:** 2022-11-10

**Authors:** Grażyna Czyżak-Runowska, Jacek Antoni Wójtowski, Bogusława Łęska, Sylwia Bielińska-Nowak, Jarosław Pytlewski, Ireneusz Antkowiak, Daniel Stanisławski

**Affiliations:** 1Department of Animal Breeding and Product Quality Assessment, Faculty of Veterinary Medicine and Animal Science, Poznań University of Life Science, ul. Słoneczna 1, Złotniki, 62–002 Suchy Las, Poland; 2Faculty of Chemistry, Adam Mickiewicz University in Poznań, ul. Umultowska 89b, 61–614 Poznań, Poland; 3Computer Lab, Poznań University of Life Sciences, ul. Wołyńska 33, 60–637 Poznań, Poland

**Keywords:** sheep yogurt, sheep kefir, lactose, rheological parameters, storage

## Abstract

**Simple Summary:**

The consumption of fermented milk products can have health benefits, particularly for those that suffer from lactose intolerance. Condition that is usually caused by insufficient production in the intestinal epithelium of lactase, the enzyme that facilitates the digestion of milk sugar (lactose). The lactose originally present in fermented beverages undergoes fermentation to lactic acid, so its concentration is lower than in processed milk. The aim of this study was to determine the effects of the storage time of fermented beverages made from sheep’s milk on the lactose content of kefir and yogurts inoculated with various starter cultures. The drinks were made from sheep’s milk with natural fat content. A sensory evaluation of the products was carried out to determine the most favorable date of consumption in terms of lactose concentration and organoleptic values. Effects of both inoculation type and beverage storage time were shown for all the parameters. Analysis showed that the optimal storage time for natural yogurt was 21 days. If kefir is to have its optimal taste, storage for more than fourteen days is not advised.

**Abstract:**

The aim of the research was to evaluate lactose content and rheological, physical, chemical, and organoleptic parameters during the storage of fermented beverages made from sheep’s milk. The research was carried out on natural, probiotic, and Greek-type yogurts, as well as kefir. The products were made using the thermostat method from the milk of 42 East Frisian sheep in the middle lactation period, in duplicate. Lactose contents, active and titratable acidity, color by the L*a*b*C*h* system, and rheological parameters (hardness, consistency, consistency, and viscosity) were tested, and organoleptic assessments were carried out on the first, seventh, fourteenth, and twenty-first days of storing the drinks at 4 °C. Of all drinks, the highest reduction in lactose after 21 days of storage was found to occur in kefir (52% reduction) and, among the yogurts, in the Greek yogurt (41% reduction). The product with the lowest lactose content, regardless of the storage period, was kefir. This indicates that kefir is more suitable than yogurt for people with partial lactose intolerance. Effects of both inoculation type and beverage storage time were shown to exist for all parameters. It was also found that kefirs suffered deterioration in most rheological parameters and, in general organoleptic evaluation in the final period of storage. Based on our analysis, the optimal storage time for natural yogurts and sheep’s milk kefirs at 4 °C was 21 and 14 days, respectively.

## 1. Introduction

Healthy human nutrition largely demands natural food. Sheep’s milk enjoys significant popularity and has greater nutritional and biological values than cow’s milk [1]. After cattle, sheep are the most important species for milk and dairy production in Europe, which is usually intended for cheese manufacturing [2]. In 2018, the world production of sheep’s milk was 10.37 Mt, and according to forecasts, by 2030, it is to increase by over 25% [3]. With regards to global milk production, sheep milk represents only a small portion of this sector, 1.3% [3]. Ewe milk contains more calcium, protein (including casein), fat, as well as significant amounts of conjugated linolenic acid, which is valuable for health [4]. It is additionally rich in magnesium and zinc, as well as in B vitamins and vitamins A, E, and C [5]. Shrestha et al. [6] showed that sheep’s milk, despite its higher energy and nutrient content, does not lead to significant undesirable digestive symptoms to the same extent as cow’s milk. Due to its favorable physical, chemical, and nutritional properties, it can serve as a valuable raw material for the production of fermented beverages, which are the basis of the functional food market [7,8,9]. These products have a therapeutic effect and are very popular among consumers, including people with limited tolerance for lactose. This is very important, as over 60% of the human population has a reduced ability to digest lactose due to low levels of lactase enzymatic activity [10]. In Europe, the number of people with lactose intolerance ranges from a few percent to 50%, and in Poland is about 37% [11,12].

The most popular fermented beverages in Poland are yogurt and kefir [13,14]. Their production involves controlled fermentation that begins with the addition, under appropriate conditions, of specific microflora to milk: these microflorae are lactic acid bacteria in the case of yogurt, and lactic acid bacteria and yeast for kefir. Fermentation of the basic milk sugar (lactose) lowers the pH of the milk, and proteins coagulate as a result of lactic acid formation [15]. In addition, the lactic acid in fermented beverages prevents the growth of putrefactive bacteria in the intestines, supports intestinal peristalsis, stimulates the secretion of digestive juices, supports the immune system, and—by increasing the absorption of calcium—has a protective function against osteoporosis. According to Wichrowska and Wojdyła [16], the changes that take place during the fermentation of milk allow humans to digest the resulting yogurt three times faster than unfermented milk. In addition, the lactic fermentation of dairy products results in an increase in the digestibility of proteins and thus in the number of free amino acids; β-galactosidase is produced, which intensifies the digestion of lactose in the gastrointestinal tract. Lactose is a reducing disaccharide consisting of a D-galactose molecule and a D-glucose molecule linked by β-1,4-glycosidic bonds [17]. It is also a natural prebiotic [18]. It is less sweet than glucose and sucrose and has a lower glycemic index [19].

Huppertz and Gazi [20] indicate that lactose plays an important role in shaping the nutritional properties and technical usefulness of milk. In fermented milk drinks, lactose is fermented to lactic acid, leaving it at levels 10–20% lower than in processed milk. It should also be emphasized that the consumption of fermented milk drinks introduces bacterial lactase to the human body; this enzyme additionally supports the digestion of lactose. Hanh et al. [21] found that almost all proteins contained in sheep’s yogurt—unlike those in cow’s and goat’s yogurt—undergo the most intense decomposition in the human stomach in the first phase of digestion. Kefir also has many health benefits due to its antibacterial, anticancer, antidiabetic, and intestinal microflora modeling properties [22]. According to Yilmaz-Ersan et al. [23], sheep’s kefir contains more antioxidants than cow’s milk.

The aim of the present study was to determine the changes that occur in the lactose concentration of fermented beverages made from sheep’s milk—namely, natural probiotic and Greek yogurts, as well as kefir—during their 21 days of storage, as well as the parallel physicochemical, rheological, and organoleptic changes. The literature lacks comprehensive studies on the effects of the storage period of fermented beverages made from sheep’s milk on the level of lactose in the context of their potential use as functional foods. An equally important aspect of the research was to assess the suitability of starter cultures intended for the production of fermented beverages from cow’s milk in the production of sheep’s milk products.

## 2. Materials and Methods

### 2.1. Milk for Producing Fermented Beverages 

The study was carried out on natural yogurt (NY), probiotic yogurt (PY), Greek yogurt (GY), and kefir (K) produced under laboratory conditions from sheep’s milk with its natural fat content, using the thermostat method. The products were produced in duplicate in the month of May, after an interval of two weeks from the collection of the bulk milk in morning milkings of 42 dairy ewes (Polish dairy sheep line 05, with a genetic share of 3/16 Polish Merino and 13/16 East Friesian [24]). The ewes (body weight of 61.0 ± 2.5 kg) were kept in group pens and fed 1 kg concentrate and 2.4 kg alfalfa/grass silage and meadow hay (dry matter basis) per ewe per day. The ewes had free access to water and a mineral saltlick. The diets were formulated to meet the animal’s nutrient requirements: 1.61 UFM (unit for milk production) and 157 g PDI (protein truly digestible in the small intestine) [25].

Milk was obtained in the milking parlor using a pipeline milking machine (Polanes, Bydgoszcz, Poland) with pre-milking and post-milking disinfection of udders and teats [26]. Before milking, the udders and teats of the sheep were cleaned using wet udder paper towels with natural cleansing ingredients (Biocell™, Delaval, Tumba, Sweden). Post-milking dipping was performed by dipping the teats in a 20% solution of iodine agent Dipal™ Conc (Delaval, Tumba, Sweden).

The basic chemical composition of the milk, presented in Table 1, was determined using Dairy Spec FT, Bentley Instruments (Chaska, MN, USA). This device uses an industrial Fourier Transform Spectrometer (FTIR) that captures the complete infrared absorption spectrum of the milk sample for component analysis.

The number of somatic cells in the milk was determined as 36.00 × 10^3^ cells mL^−1^ (Bacto Count IBCm, Bentley Instruments, MN, Chaska, USA).

### 2.2. Production of Yogurt and Kefir Samples

To produce the fermented beverages, ten liters of milk were pasteurized at a time at 75 °C for 30 min (FJ 15 Eco mini pasteurizer, Milky, Althofen, Austria). Then, for the yogurts, the milk was cooled to 43 °C and inoculated with the appropriate starter culture at 1% by volume of milk; the following inoculants were used:-Natural yogurt: YO 122 inoculant from Biochem (*Streptococcus salivarius* subsp. *thermophilus*, *Lactobacillus delbrueckii* subsp. *bulgaricus*);-Probiotic yogurt: ABY inoculant from Biochem (*Streptococcus salivarius* subsp. *thermophilus*, *Bifidobacterium bifidum*, *Lactobacillus acidophilus*, *Lactobacillus delbrueckii* subsp. *bulgaricus*);-Greek yogurt: Lyofast Y 480F inoculant from Sacco (*Lactobacillus delbrueckii* ssp. *bulgaricus*, *Streptococcus thermophilus*).

The inoculated milk was poured into 100 mL sterile unit packages and incubated at 43 °C until set (Nuve EN 055 Incubator, Ankara, Turkey). The products were then cooled to 4 °C. They were qualitatively evaluated after 24 h (fresh: day 1). This analysis was repeated on days 7, 14, and 21 of storage at 4 °C. The lactose content, active acidity (pH), titratable acidity (potential), L*a*b*C*h* color, and rheological parameters (hardness, consistency, consistency, and viscosity) were analyzed, and an organoleptic evaluation was performed.

To produce the kefir, the pasteurized milk was cooled to room temperature (around 23 °C) and inoculated with LYOFAST MT 036LX from Sacco (*Lactococcus lactis* ssp. *lactis, Lactococcus lactis* ssp. *cremonis*, *Lactococcus lactis* ssp. *diacetylactis, Lactobacillus brevis*, *Leuconostoc*, and the yeasts *Saccharomyces cerevisiae*, *Kluyveromyces lactis*). It was then incubated for about 18 h at inoculation temperature until set before being cooled to 4 °C.

After 24 h, and then on days 7, 14, and 21 days of refrigerated storage, the same quality parameters were measured as in the case of the yogurts. 

### 2.3. Physical and Chemical Analysis 

The lactose content of the products was determined in accordance with the method described by Polish Standard PN-68/A-86122 [27], which is specified as a suitable method for evaluating milk samples and dairy products by the Polish Ministry of Rural Development’s Regulation of 12 December 2002 (Dz.U. 2002.230.1931). This method involves determining the amount of halogen which has been reduced by the reaction of lactose in the tested sample with chloramine T and potassium iodide. The individual steps of the analytical procedure are described in the supplementary materials to this article (Appendix A).

The pH was determined using a Schotte Handylab 2 pH meter with a Schott L68880 glass-calomel electrode (Schotte, Mainz, Germany). Standard buffers at pH 4.01 and 7.00 were used for calibration (Chempur, Piekary Śląskie, Poland). 

In order to determine the potential acidity, titration with 0.25 M NaOH was used using the Soxhlet–Henkel method [27].

Rheological parameters were determined using a TA-TX Plus micro stable texture analyzer (Micro Stable Micro Systems, Golborne, Warrington, UK). A penetration test was performed with a penetration rate of 1 mm/sec and a probe penetration depth of 30 mm. The 100 mL sample was analyzed in a plastic cup with a diameter of 55 mm. The analysis employed a flat probe with a diameter of 35 mm, intended for testing liquid, semiliquid, and viscous substances. This was placed centrally over the container with the product sample. Hardness (g), consistency (g/s), cohesiveness (g), and viscosity index (gs) were measured. 

The color was measured using a Konica Minolta model CR5 desktop colorimeter (Minolta, Osaka, Japan). This was calibrated to the following parameters: observer 2°, illuminant C. Yogurt samples of about 50 mL at room temperature were placed in a special spectrophotometric dish (CR-A504 diameter 34/35), and the space was tested with the CIE L*a*b*C*h* system; here L* refers to lightness, a* to the proportion of red in the spectrum, b* to the proportion of yellow in the spectrum, C* to saturation, and h* to hue [28]. 

All chemical analyses were performed in triplicate, and the rheological parameters were determined in five replications, from which the arithmetic mean was calculated.

### 2.4. Sensory Examination

A trained sensory panel assessed the coded beverage samples at random, following the methodology described by Wichrowska and Wojdyła [16]. All panel members (*n* = 10, six females and four males aged 35–58) had been trained in sensory examination [29,30]. Samples were analyzed 24 h after the end of the fermentation process and sequentially on days 7, 14, and 21 of storage. The samples in airtight polystyrene containers were conditioned at room temperature for ten minutes before testing. The samples were scored on a five-point scale. Water was provided to rinse the palate before and after tasting. The test was conducted in triplicate.

### 2.5. Statistical Analysis 

The data were analyzed using a fully randomized, repeated-measures design using the Mixed procedure in SAS, v 9.4 (SAS Institute, Cary, NC, USA) with the following linear model: Y_ijk_ = µ + p_i_ + t_j_ + (pt)_ij_ + e_ijk_

where Y_ijk_ is the mean of observation, μ the overall mean, p_i_ the constant effect of the type of dairy product (i.e., the constant effect of storage time), and the (pt)_ij_ the interactions of product type × time. The VC covariance matrix (variance components) was used; this was determined using the Akaike information criteria. The significance of the differences between the object means was determined using the adjusted Tukey’s test at the significance level *p* ≤ 0.05. 

Linear trends in the concentration of lactose during storage were determined using the second-degree linear regression method; the R^2^ determination coefficients of the linear equation were also estimated. 

## 3. Results

Table 2 shows the results for the changes in lactose concentration in the fermented milk beverages, depending on time stored at 4 °C. Statistical analysis revealed significant relationships (*p* < 0.0001) between storage time and type of fermented beverage on one hand and lactose content on the other. A statistically highly significant interaction of beverage type × storage time was also demonstrated (*p* < 0.0001).

A statistically significant decrease in lactose content was found with the extension of the storage period of fermented milk products. During the 21-day storage period, the greatest reduction in lactose was seen in kefir (52%) and Greek yogurt (41%). In the natural and probiotic yogurts, the reduction was much lower and similar (26–29%). When the lactose content in the products was compared on the same day of their storage, the lowest amount was found in kefir. This product differed significantly in terms of lactose content from the other fermented milk beverages. The estimated linear trends in lactose content over time had high coefficients of determination R^2^, confirming the accuracy of the estimated value (Figure 1). The linear trend calculated for kefir had the highest value of R^2^ = 0.9504.

Table 3 presents the results regarding the effects of storage time and product type on pH, titratable acidity, rheological properties, and color of the fermented milk beverages. Statistical analysis showed a significant influence of the type of product and its storage time on all the properties of fermented milk beverages. In all features, except for the rheological parameters, significant interactions of beverage type × storage time were found. The lowest pH values were seen in all the yogurts on days 14 and 21 days of storage. The pH of the natural and probiotic yogurts on days 14 and 21 differed significantly (*p* < 0.05) from the values for the previous days. There was no effect of storage time on the pH of the kefir (*p* > 0.05). The pH value of this product on the first and second measurement dates was the lowest among all tested fermented beverages. 

Kefir showed the highest titratable acidity, except for GY, on day 7 of storage. It has been shown that the titratable acidity of the products increases with the extension of the storage period. Greek yogurt had the greatest acidity (63.87° SH); this was followed by kefir (52.93° SH). These values significantly differed from the acidity of these beverages earlier in the storage period (*p* < 0.05). There were no significant differences between the natural and probiotic yogurts in titratable acidity on days 14 and 21 of storage (*p* > 0.05). On the other hand, when comparing the acidity of beverages on the same day of storage, the highest statistically significant acidity was found in Greek yogurt on days 14 and 21. 

The kefir was distinguished by the lowest values of curd hardness on the individual test dates (Table 4). The kefir’s greatest hardness of 63.60 g was found on day 14, while the smallest hardness value of 45.08 g was seen on day 21 of storage (*p* < 0.05). The greatest hardness of all the products was found for the natural and probiotic yogurts. 

Only in the case of kefir was firmness significantly determined by storage time (*p* < 0.05); The lowest firmness was found on day 21 of storage. This hardness value differed significantly from those of the other products during storage (*p* < 0.05). The kefir was significantly less firm on days 7, 14, and 21 of storage than were the yogurts (*p* < 0.05). On the other hand, the firmness of the yogurts did not change significantly over the entire storage period (*p* < 0.05), except in the case of the natural and probiotic yogurts on day 7 (*p* < 0.05). 

Of all the fermented beverages, the highest value of cohesiveness during storage was displayed by kefir (*p* < 0.05). All the fermented milk drinks reached their highest cohesiveness values on day 21 of storage.

The natural yogurt reached its statistically significantly lowest viscosity on day 7, and the Greek yogurt and kefir did so on day 21 of storage. The kefir had the highest viscosity index of all the fermented milk beverages at all times (α = 0.05).

When examining color components, we found that the fermented milk drinks differed significantly from each other in terms of the brightness L* on the day of production and on day 21 of storage (*p* < 0.05). On those days, the Greek yogurt was the lightest (91.94) and the least light (90.83) of all the dairy products. In all the analyses, the proportion of a* color was negative (representing green), while the proportion of b* color was positive (representing yellow). The effect of storage time on this parameter was observed only in the case of kefir, for which the highest proportion of green (4.30) and yellow (13.02) colors was recorded on day 21 of storage. The probiotic and natural yogurts had the highest value of the a* color level (*p* < 0.05). The lowest values of the b* color level were observed for kefir on days 1, 7, and 14 of its storage (*p* < 0.05). 

The color saturation C* of the kefir increased with storage time. Additionally, until day 14 of storage, the kefir had the lowest color saturation of all the fermented milk beverages (*p* < 0.05). 

During the entire storage period, the natural and probiotic yogurts (*p* < 0.05) showed the highest h* values for this parameter.

The effects of storage time and the type of fermented beverage on selected organoleptic characteristics of the product are presented in Table 4 and Figure 2. We found that there was a statistically significant influence of storage time on the color, texture, and aroma of the beverages (*p* < 0.001). In each of the time intervals, significant differences were found in the color of both probiotic yogurt and kefir (*p* < 0.05). The lowest values of this parameter of any day of storage were observed on day 21 (*p* < 0.05). 

NY achieved the highest overall score at all assessed time periods and was ahead of PY in this respect. The kefir’s scores (K) for this trait on days 1 to 14 were good and remained relatively stable before deteriorating sharply on day 21 of storage. The score for K on day 21 of storage was the lowest of all products (*p* < 0.05). It should be emphasized that the overall assessment of the GY was relatively low and was the lowest among all the products for the first three time intervals.

## 4. Discussion

Sheep’s milk’s rich chemical composition makes it an excellent raw material for the production of fermented beverages and matured cheeses. Lactose (milk sugar) is a basic component of milk, and its presence has a significant effect on the processing and stability of dairy products. The lactose content of sheep’s milk is variable and, according to Pandya and Ghodke [31], ranges from about 4.2 to 5.4%. On the other hand, Barłowska et al. [32] determined the average concentration of lactose in sheep’s milk to be 4.75%. This research shows that the amount of lactose in fermented products decreased with the storage period. This process involves the consumption of lactose by individual microorganisms (bacteria and yeasts) during the ongoing lactic and alcoholic fermentation. However, given the lactose concentrations available in each storage period, it can be concluded that the intensity of lactic fermentations taking place in the individual fermented sheep products was different. 

In this study, kefir had the lowest lactose content. These results demonstrate that kefir underwent a more intense reduction of lactose the longer its storage period was extended. The intensity of this process is associated with the presence of specific types of bacterial cultures and yeasts. It is thought that when fermented beverages are consumed, the digestion of lactose is stimulated because the lactic acid bacteria contained in kefir or yogurt perform the same functions as the lactase enzyme, hydrolyzing the β-1,4-glycosidic bond both during the lactic acid fermentation process that produces the milk drink and in the digestive tract after consumption. According to Montalto et al. [33], lactic acid fermentation reduces the concentration of lactose in the product by 25–50%. Among the fermented sheep drinks assessed in our present research, kefir stands out for having the lowest concentration of milk sugar on day 21 of storage at 4 °C. The reduction of lactose in this kefir was 52% in relation to the concentration of milk sugar on the day of production. 

The results suggest that kefir made from sheep’s milk has exceptional importance as a functional food. It could be used in the diet of patients with hypolactasia, as these patients do not require the complete elimination of dairy products from their diet but only a reduction that depends on the individual’s changing needs [34].

Significant factors that affect the coagulation of milk in the production of fermented beverages are its initial pH and the amounts of lactose and casein it contains [35]. De Morais et al. [36] observed a decrease in pH during the storage of goat’s milk yogurt, which was caused by the continuous production of lactic acid by lactic acid bacteria. Similarly, our research showed a decrease in pH in the sheep milk yogurts with the passage of time from production. However, the value of this parameter on days 14 and 21 of storage at 4 °C was not statistically significant. 

In the study of Wichrowska and Wojdyła [16], carried out on natural yogurts made from cow’s milk and organic cow’s milk, a continuous decrease in the acidity of the dairy products was found until day 14 of storage. Bierzuńska et al. [37], in their analysis of yogurts made with the addition of polymerized whey proteins, found a decrease in acidity only after 21 days of storage. Most likely, the increase in the acidity of the milk produced during the storage period is determined by the increasing activity of the microorganisms that ferment lactose and produce lactic acid [7,8]. On the other hand, the time during which the decrease in pH is halted is due to the depletion of sugars as the main food for microorganisms. According to Gaspar et al. [38], at that time, proteins are digested, and metabolites are produced, which raises the pH level. Our research also looked at the titratable (potential) acidity of fermented milk drinks during their storage period. This parameter describes the concentration of all acidic chemicals in the product. Changes in titratable acidity during the storage period of the milk products were consistent with the development of active acidity. After day 21 of storage, the product with the highest titratable acidity turned out to be the Greek yogurt (63.87° SH); this was significantly lower for the kefir (52.93° SH). Our research shows that titratable acidity is affected by both the type of inoculant used, the storage time of the products, and the interaction that occurs. 

Assessment of the rheological properties of fermented milk beverages (their hardness, firmness, cohesiveness, and viscosity) plays an important role in assessing the quality of the products and their compliance with storage requirements. The rheological properties of yogurts are affected by the production technology and additives used, as well as by the type of starter cultures and their activity. *Lactobacillus delbrueckii* ssp. *bulgaricus* and *Streptococcus thermophilus* bacterial strains are used in the production of yogurts. These are capable of synthesizing exopolysaccharides (EPS), which improve the physical stability of natural yogurts [39]. Examples of EPS produced by lactic acid bacteria include dextran, alternan, mutan, reuteran, xanthan, gellan, pullulan, and alginia. They are secreted outside the bacterial cell in the form of mucus or are attached to the cell surface [40]. In our research, the kefir on day 21 after its production had the lowest hardness and compactness of all the fermented milk drinks. The yogurt varieties we examined showed the greatest hardness and firmness on day 14 of storage, although no significant statistical differences were noted between the averages obtained in the individual evaluation periods. Nguyen et al. [41] indicated that yogurts made from sheep’s milk have high hardness due to the high concentration of protein in this milk, compared to that of other animal species. On the other hand, another factor that affects the hardness of the product may be the type of microorganism causing the fermentation process. The study of Costa et al. [42] showed that the higher hardness of fermented milk drinks derived from goat’s milk was caused by the presence of Lactobacillus delbrueckii ssp. bulgaricus and Lactobacillus acidophilus. The cohesiveness of the milk drink indicates a strong aggregation of its molecules, making the texture becomes stronger. In our own research, kefir had higher cohesiveness and viscosity than yogurts throughout the entire period of storage (21 days). The viscosity changed during the storage of all dairy products except for the probiotic yogurt. The highest viscosity was found for the remaining fermented milk drinks other than natural yogurt on day 21 of storage at 4 °C. The increase in yogurt viscosity with the length of its storage may be due to the decrease in pH. The research of Bierzuńska et al. [37] on yogurts and kefirs made from cow’s milk showed that the higher the concentration of whey proteins in relation to casein proteins, the less compact and cohesive the kefir was; it also had a thinner consistency and a lower viscosity index than yogurt. Shihata and Shah [43] found that greater hardness and viscosity index for fermented milk beverages is associated with the attachment of mucinogenic strains to the protein matrix with exopolysaccharides. In our research, when we examined the color components of the fermented sheep milk beverages, no statistically significant change in their brightness (L*) was found over the 21 days of the storage period. Jakubowska and Karamucki obtained similar results for retailed cow’s milk yogurts [44]. When the fermented milk drinks in our study were compared on a specific storage day, it was found that Greek-type yogurt had the highest value of this parameter just after production and the lowest on day 21. Teichert et al. [45], on the basis of their research on fermented goat’s milk products, suggested that color and acidity changes are the result of the continuous degradation of lactose by bacteria contained in cultures that were used. In those authors’ study, an increase in color lightness (L*) was found in goat yogurt on day 21 of refrigerated storage. In our research, the color parameters a*, b*, and C*, were significantly affected by the storage period only for kefir: as the storage time increased, stronger color saturation and a shift towards green and yellow were observed. Kefir had the greatest values for these parameters on day 21 of storage. However, on individual days of the storage period, kefir showed the lowest values of the color parameters a*, b*, and C* compared to the yogurts. For Greek yogurt, the storage time had significance for the color hue (h*) parameter, the lowest values of which were obtained on days 14 and 21 of storage. Cais-Sokolińska and Majcher [46] indicate that the change in the color of milk fermented beverages during a storage period results from a change in the casein complex from micellar to dispersion.

Table 3 and Figure 2 present the results of the organoleptic evaluation of the four fermented sheep’s milk products during the period of storage at 4 °C. They show the deterioration of the parameters with the extension of the storage period. Statistical analysis showed the statistically significant lowest scores on day 21 of storage for the color of probiotic yogurt and kefir, while kefir had the lowest score for aroma. Our results show the relatively high variability of the parameters, particularly in the case of kefir. The lower quality of fermented milk products that results from the extension of the storage period is most likely due to metabolic processes affecting proteins, carbohydrates, and fats. The use of bacterial starter cultures is also significant. Moreover, Marii et al. [47] indicate the possibility of spoilage of sheep yogurts during storage as a result of the growth of microorganisms, including yeasts and molds. 

## 5. Conclusions

The results of the research show that the organoleptic parameters of the fermented sheep beverages deteriorated during storage, with kefir showing the greatest decrease in its scores; this may indicate lower suitability of this product for longer storage at 4 °C. The test results showed that the starter cultures YO 122 and ABY (Biochem, Monterotondo-Rome, Italy), as well as LYOFAST Y 480F and LYOFAST MT 036LX (Sacco, Kobylniki-Kościan, Poland), were fully suitable for the production of fermented beverages from sheep’s milk.

The level of lactose and the results of the organoleptic evaluation of the finished products showed that, for the tested sheep’s milk fermented beverages, the optimal storage time at 4 °C was 14 days for kefir and 21 days for the yogurts, regardless of the starter culture used. 

Our results on lactose reduction allow us to make the case that kefir made from sheep’s milk has exceptional importance as a functional food. Kefir could find application in the diet of patients with hypolactasia, as such patients do not require the complete elimination of dairy products from their diet, but only a reduction that depends on the individual’s changing needs.

## Figures and Tables

**Figure 1 animals-12-03105-f001:**
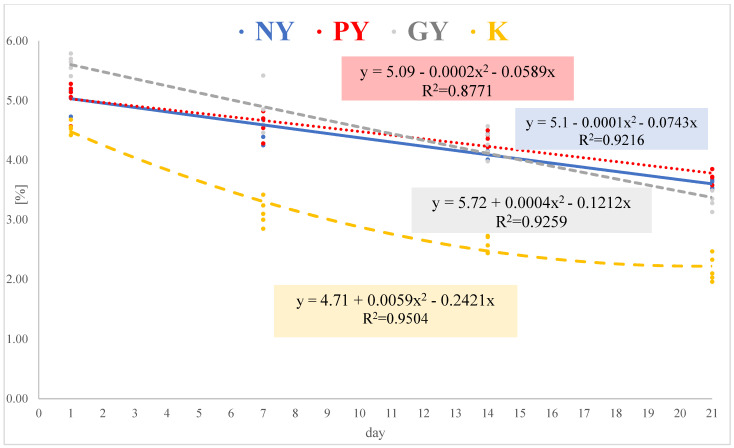
Linear trends in the concentration of lactose during storage. R^2^—determination coefficients of the linear equation. NY—Natural yogurt, PY—Probiotic yogurt, GY—Greek yogurt, and K—Kefir.

**Figure 2 animals-12-03105-f002:**
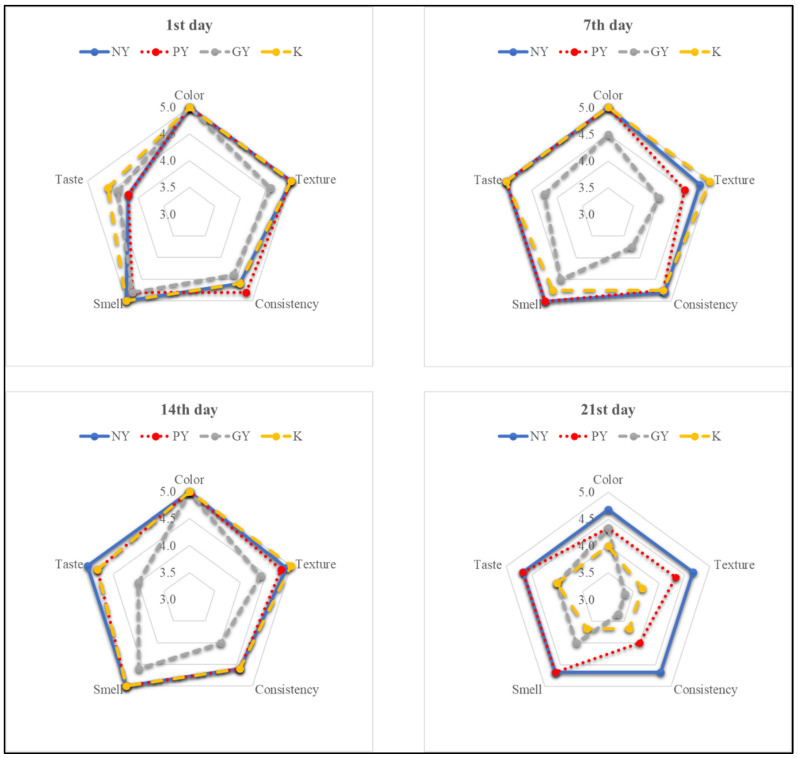
The effects of storage time and the type of fermented beverage on selected organoleptic characteristics of the product. NY—Natural yogurt, PY—Probiotic yogurt, GY—Greek yogurt, and K—Kefir.

**Table 1 animals-12-03105-t001:** Chemical composition of bulk tank milk ($\bar{x}\;$ ± SD).

Traits	$\bar{x}\;\;\pm \;\mathrm{SD}$
Dry matter (%)	15.74 ± 0.18
Protein (%)	3.87 ± 0.02
Fat (%)	5.38 ± 0.01
Lactose (%)	5.64 ± 0.01
Mineral salts (%)	0.72 ± 0.01

$\bar{x}\;$—mean value; SD—standard deviation.

**Table 2 animals-12-03105-t002:** Lactose content (%) in fermented milk drinks depends on the storage time (at 4 °C) and the type of milk drink.

Milk Drink	Storage Time (Days)				Se	Type of Drink	Time	Type of Drink × Time
	1	7	14	21		*p*-Value		
NY PY GY K	5.05 ^aA^	4.54 ^bA^	4.14 ^cA^	3.59 ^dA^	0.08	0.0001	0.0001	0.0001
	5.05 ^aA^	4.60 ^bA^	4.29 ^cA^	3.76 ^dA^				
	5.65 ^aB^	4.77 ^bA^	4.23 ^cA^	3.34 ^dB^				
	4.55 ^aC^	3.12 ^bB^	2.65 ^cB^	2.17 ^dC^				

^a–d^ Values in rows with different lowercase letters are significantly different (*p* < 0.05). ^A–D^ Values in columns with different uppercase letters are significantly different (*p* < 0.05). NY—Natural yogurt, PY—Probiotic yogurt, GY—Greek yogurt, and K—Kefir.

**Table 3 animals-12-03105-t003:** Effect of storage time (at 4 °C) and type of drink on the pH value, acidity, rheological properties, and color of fermented milk beverages.

Traits	Milk Drink	Storage Time (Days)				Se	Type of Drink	Time	Type of Drink × Time
		1	7	14	21		*p*-Value		
pH	NY	4.69 ^aA^	4.47 ^bA^	4.33 ^c^	4.36 ^cAB^	0.02	0.0001	0.0001	0.0005
	PY	4.66 ^aA^	4.53 ^bA^	4.38 ^b^	4.48 ^bA^				
	GY	4.38 ^aAB^	4.77 ^aB^	4.17 ^b^	4.23 ^abB^				
	K	4.26 ^B^	4.33 ^C^	4.26	4.36 ^AB^				
Acidity (°SH)	NY	36.93 ^aA^	40.53 ^aA^	45.47 ^bA^	47.60 ^bA^	1.48	0.0001	0.0001	0.0001
	PY	36.67 ^aA^	42.00 ^abA^	46.53 ^bAB^	48.93 ^bA^				
	GY	40.40 ^aA^	45.73 ^bAB^	53.47 ^cC^	63.87 ^dB^				
	K	47.33 ^aB^	48.40 ^aB^	49.73 ^aB^	52.93 ^bC^				
Hardness (g)	NY	135.29	145.33 ^A^	167.19	134.32 ^A^	7.12	0.0001	0.0441	0.9587
	PY	114.53	134.12 ^A^	172.10	128.99 ^AB^				
	GY	95.62	86.22 ^B^	167.56	102.92 ^AB^				
	K	48.48 ^b^	47.55 ^abB^	63.60 ^a^	45.08 ^bB^				
Consistency (g/s)	NY	2964.07	3404.31 ^A^	3446.80 ^A^	2849.52 ^A^	14.77	0.0001	0.0468	0.8971
	PY	2281.40	2576.18 ^AB^	3653.89 ^A^	2317.79 ^A^				
	GY	2020.62	1946.75 ^BC^	3392.86 ^A^	2096.91 ^A^				
	K	1229.04 ^a^	1218.43 ^abC^	1431.58 ^aB^	1112.74 ^bB^				
Cohesion (g)	NY	−154.32 ^A^	−135.77 ^A^	−196.77 ^A^	−125.85 ^A^	8.99	0.0001	0.0258	0.8951
	PY	−138.48 ^A^	−144.96 ^B^	−167.70 ^B^	−109.57 ^A^				
	GY	−155.52 ^A^	−65.45 ^C^	−145.07 ^B^	−59.61 ^B^				
	K	−22.27 ^C^	−19.33 ^D^	−34.86 ^C^	−17.08 ^B^				
Viscosity (g/s)	NY	−290.04 ^aA^	−104.98 ^bA^	−286.28 ^aA^	−217.03 ^abA^	13.62	0.0001	0.0001	0.0946
	PY	−248.43 ^A^	−236.01 ^B^	−286.27 ^A^	−180.80 ^A^				
	GY	−180.51 ^aAB^	−96.73 ^abA^	−174.61 ^abA^	−59.46 ^bB^				
	K	−48.53 ^aB^	−29.16 ^abC^	−58.07 ^aB^	−14.81 ^bB^				
L*	NY	91.60 ^AB^	91.25	91.58	91.24 ^AB^	0.04	0.0152	0.0458	0.0011
	PY	91.52 ^A^	91.51	91.68	91.68 ^A^				
	GY	91.94 ^B^	91.69	91.30	90.83 ^B^				
	K	91.51 ^A^	91.65	91.56	91.65 ^A^				
a*	NY	−4.99 ^A^	−5.12 ^A^	−5.20 ^AB^	−5.04 ^A^	0.08	0.0001	0.0023	0.0001
	PY	−5.17 ^A^	−5.05 ^A^	−5.31 ^A^	−5.19 ^A^				
	GY	−4.68 ^A^	−4.80 ^A^	−4.62 ^BC^	−4.30 ^B^				
	K	−2.80 ^aB^	−4.02 ^bB^	−4.22 ^bC^	−4.30 ^bB^				
b*	NY	13.27 ^A^	14.06 ^A^	13.94 ^AB^	13.87	0.20	0.0010	0.0001	0.0001
	PY	13.99 ^A^	13.88 ^A^	14.40 ^A^	14.27				
	GY	13.48 ^A^	14.09 ^A^	14.37 ^A^	14.04				
	K	8.21 ^aB^	11.65 ^bB^	12.55 ^bB^	13.02 ^b^				
C*	NY	14.17 ^A^	14.96 ^A^	14.88 ^A^	14.76 ^AB^	0.21	0.0010	0.0001	0.0001
	PY	14.92 ^A^	14.78 ^A^	15.34 ^A^	15.18 ^A^				
	GY	14.27 ^A^	14.89 ^A^	15.10 ^A^	14.69 ^AB^				
	K	8.68 ^aB^	12.32 ^bB^	13.24 ^bB^	13.72 ^bB^				
h*	NY	110.59 ^A^	109.99 ^A^	110.45 ^A^	109.95 ^A^	0.13	0.0010	0.0011	0.0490
	PY	110.27 ^AB^	109.99 ^A^	110.23 ^A^	109.99 ^A^				
	GY	109.15 ^aBC^	108.82 ^aB^	107.84 ^abB^	107.09 ^bB^				
	K	108.84	108.99 ^B^	108.59 ^B^	108.27 ^B^				

^a–d^ Values in rows with different lowercase letters are significantly different (*p* < 0.05). ^A–D^ Values in columns with different uppercase letters are significantly different (*p* < 0.05). NY—Natural yogurt, PY—Probiotic yogurt, GY—Greek yogurt, K—Kefir, L*—brightness (values from 0 to 100), a*—share of green and red colors (green—negative value and red—positive value), b*—share of blue and yellow colors (blue—negative value and yellow—positive value), C*—saturation, and h *—hue.

**Table 4 animals-12-03105-t004:** The impact of storage time (at 4 °C) and type of drink on selected organoleptic characteristics of fermented milk beverages (points).

Traits	Milk Drink	Storage Time (Days)				Se	Type of Drink	Time	Type of Drink × Time
		1	7	14	21		*p*-Value		
Color	NY	5.00	5.00	5.00	4.67	0.05	0.2220	0.0001	0.2473
	PY	5.00 ^a^	5.00 ^a^	5.00 ^a^	4.33 ^b^				
	GY	5.00	4.50	5.00	4.33				
	K	5.00 ^a^	5.00 ^a^	5.00 ^a^	4.00 ^b^				
Texture	NY	5.00	4.80	4.90	4.67 ^A^	0,08	0.0013	0.0003	0.5392
	PY	5.00	4.50	4.80	4.33 ^A^				
	GY	4.60	4.00	4.40	3.33 ^B^				
	K	5.00 ^a^	5.00 ^a^	5.00 ^a^	3.67 ^bB^				
Consistency	NY	4.60	4.80	4.60	4.67	0.10	0.0367	0.1343	0.9201
	PY	4.80	4.75	4.60	4.00				
	GY	4.40	3.75	4.00	3.33				
	K	4.60	4.75	4.60	3.67				
Smell	NY	5.00	5.00	5.00	4.67 ^A^	0.06	0.0065	0.0001	0.2304
	PY	4.80	5.00	5.00	4.67 ^A^				
	GY	4.80	4.50	4.60	4.00 ^B^				
	K	5.00 ^a^	4.75 ^a^	5.00 ^a^	3.67 ^bB^				
Taste	NY	4.20	5.00	5.00	4.67	0.09	0.1215	0.1704	0.6820
	PY	4.20	5.00	4.80	4.67				
	GY	4.40	4.25	4.00	4.00				
	K	4.60	5.00	4.80	4.00				
Overall assessment	NY	4.80	4.90	5.00	4.67 ^A^	0.07	0.0026	0.0023	0.6573
	PY	4.70	4.88	4.90	4.33 ^A^				
	GY	4.50	4.25	4.30	3.83 ^B^				
	K	4.80 ^a^	4.75 ^a^	4.90 ^a^	3.67 ^bB^				

^a–d^ Values in rows with different lowercase letters are significantly different (*p* < 0.05). ^A–D^ Values in columns with different uppercase letters are significantly different (*p* < 0.05). NY—Natural yogurt, PY—Probiotic yogurt, GY—Greek yogurt, and K—Kefir.

## Data Availability

The data presented in this study are available on request from the corresponding author.

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
