# Peer review of "Lactose Content and Selected Quality Parameters of Sheep Milk Fermented Beverages during Storage"

_animals, 2022, doi:10.3390/ani12223105_

Round 1
Reviewer 1 Report
Dear Authors,
I have carefully reviewed the manuscript Animals-1962295, entitled “Effects of the storage period of sheep milk fermented beverages on lactose content and selected quality parameters". The subject has a considerable interest for the field, as the dairy sector seeks alternatives to cow milk beverages, with high and functional nutritional properties, specially as there is a progressive increase of population with a limited ability to process lactose. The authors have done a good effort and the manuscript is in general easy to read and to understand. However, there is still some room for improvement. Please see some comments and queries below, which I hope you may find useful.
Regards.
INTRODUCTION
L44-45: Please support this sentence with some numbers regarding consumption of sheep milk. Authors state throughout the introduction that sheep milk is very popular, although more than 95% of milk from small ruminants is intended for cheesemaking. There might be a small proportion of goat milk consumed in fresh form in Europe, but the popularity of sheep fresh milk as a substitute of milk from dairy cows is, to my understanding, questionable. Perhaps the authors refer particularly to the consumer habits in Poland. Could you please elaborate?
MATERIAL AND METHODS
L100-103: Is there any particular reason for using three breeds of sheep? I understand in the end, beverages were made from bulk tank milk and there is no use whatsoever of the the different breeds in the study. I reckon that, considering the experimental design, it would have been interesting to check if there is a breed effect on the production and storage of each beverage.
L109-110: Please state here the disinfectants used for pre and post dipping. Did you use iodine dip, chlorhexidine, lactic acid, chlorine dioxide or any other? Was the formula applied in spray or foam?
L110-112: Authors cite “AOAC Oficial Methods of Analysis”, although they state that analysis of milk composition was performed on a Dairy Spec FT. This devices is based on Fourier-transform infrared spectroscopy and does not provide any chemical analysis regarded as a standard. This has really no effect on the outcomes of the paper, but authors should describe their methodology correctly.
L115-126: Species names in latin must appear in italics.
L165-169: Please cite here the international ISO guidelines for the measurement of color using the CIELab color space: CIE (2019). Colorimetry - Part 4: CIE 1976 L*a*b* colour space. 11664–4:2019. International Organization for Standardization, Geneva, Switzerland; Commission internationale de l'éclairage, Vienna, Austria.
RESULTS
Again, as stated before for L100-103, I believe that it would have been interesting to check if there is an existing breed effect on production and storage of each beverage. I think that the methodological approach completely allowed this to be done and authors have not fully taken advantage of their own experimental design. Taking this into consideration, stronger outcomes could have been obtained with no additional effort and discussion would have been richer.
DISCUSSION
The first paragraphs of this section repeat concepts that have already been introduced and described in detail in the introduction, making some parts parts of the discussion section highly redundant. Please consider rewriting or removing some of these sentences either here or in the introduction section.
CONCLUSIONS
L422-429: This paragraph only states results rather than conclusions and should not be included here. The Conclusions section needs to provide insights about the outcomes of the study and not just provide again the same data from the results section.
Author Response
REPLY TO REVIEWER 1
AU: We wish to thank you very much for the very thorough and constructive review, detailed, critical comments and suggestions, which proved very helpful in the preparation of the corrected version of the study. All changes are marked in red in the manuscript.
The manuscript has been reviewed and edited for the English language by a native speaker of English and a copyeditor experienced in preparing academic works for the press and for journals. Attached is the Certificate of English Editing.
on behalf of all authors
Prof. Dr. Jacek Antoni Wójtowski
ORCID: https://orcid.org/0000-0002-9186-006X
Reviewer(s)' Comments to Author:
INTRODUCTION
L44-45: Please support this sentence with some numbers regarding consumption of sheep milk. The authors state throughout the introduction that sheep milk is very popular, although more than 95% of milk from small ruminants is intended for cheesemaking. There might be a small proportion of goat milk consumed in fresh form in Europe, but the popularity of sheep fresh milk as a substitute of milk from dairy cows is, to my understanding, questionable. Perhaps the authors refer particularly to the consumer habits in Poland. Could you please elaborate?
AU:
Thank you for your valuable attention. For forty years we have been dealing with the issues of obtaining and processing milk from small ruminants and we observe the trends occurring in this branch of production. Sheep's milk production has been and will be a typical niche production. Writing about the growing popularity of sheep's milk, we absolutely did not mean the consumption of sheep's milk, but the generally increasing production of this type of milk, used mainly for the production of cheese. In our opinion, specifying the consumption level of sheep's milk can be misleading, as there are no fully reliable data, mainly due to the very local nature of production and consumption. Most of the data is estimates. Taking this into account, we have completed the paragraph as follows: “Healthy human nutrition largely requires natural food. Sheep's milk is very popular and has greater nutritional and biological values than cow's milk [1]. After cattle, sheep are the most important specie for milk and dairy production in Europe, which is usually intended for cheese manufacturing (Costa et al., 2022). In 2018, the world production of sheep's milk was 10.37 Mt and according to forecasts, by 2030 it is to increase by over 25% (Pulina et al., 2018). With regards to global milk production, sheep milk represents only a small portion of this sector, 1.3 %, (Pulina et al., 2018).
MATERIAL AND METHODS
L100-103: Is there any particular reason for using three breeds of sheep? I understand in the end, beverages were made from bulk tank milk and there is no use whatsoever of the the different breeds in the study. I reckon that, considering the experimental design, it would have been interesting to check if there is a breed effect on the production and storage of each beverage.
AU:
The milk came from only one breed of sheep and it was Polish dairy sheep line 05 with a genetic share of 3/16 Polish Merino and 13/16 East Friesian. For more information on this population, see the paper: Gut, A .; Wójtowski, J .; Stanisz, M .; Ślósarz, P. Dairy performance of new Polish milk sheep. Ann. Anim. Sci. 2008, 8, 411-415. The word "genetic share" has been added to this paragraph to make it more communicative.
L109-110: Please state here the disinfectants used for pre and post dipping. Did you use iodine dip, chlorhexidine, lactic acid, chlorine dioxide or any other? Was the formula applied in spray or foam?
AU:
Milk was obtained in the milking parlor using a pipeline milking machine (Polanes, Bydgoszcz, Poland) with pre-milking and post-milking disinfection of udders and teats [26]. Before milking, the udders and teats of the sheep were cleaned using wet udder paper towels with natural cleansing ingredients (Biocell ™, Delaval, Tumba, Sweden). Post-milking dipping was performed by dipping the teats in a 20% solution of iodine agent Dipal ™ Conc (Delaval, Tumba, Sweden).
L110-112: Authors cite “AOAC Oficial Methods of Analysis”, although they state that analysis of milk composition was performed on a Dairy Spec FT. This devices is based on Fourier-transform infrared spectroscopy and does not provide any chemical analysis regarded as a standard. This has really no effect on the outcomes of the paper, but authors should describe their methodology correctly.
AU:
Thank you for your valuable and right attention. This sentence was included in the text by mistake and has already been deleted. We added: This device uses an industrial Fourier Transform Spectrometer (FTIR) that captures the complete infrared absorption spectrum of the milk sample for component analysis.
L115-126: Species names in latin must appear in italics.
AU:
Sorry, a technical error has occurred. The spelling was corrected.
L165-169: Please cite here the international ISO guidelines for the measurement of color using the CIELab color space: CIE (2019). Colorimetry - Part 4: CIE 1976 L*a*b* colour space. 11664–4:2019. International Organization for Standardization, Geneva, Switzerland; Commission internationale de l'éclairage, Vienna, Austria.
AU:
Thank you for your valuable attention. We added the citation: CIE (2019). Colorimetry - Part 4: CIE 1976 L*a*b* colour space. 11664–4:2019. International Organization for Standardization, Geneva, Switzerland; Commission internationale de l'éclairage, Vienna, Austria.
RESULTS
Again, as stated before for L100-103, I believe that it would have been interesting to check if there is an existing breed effect on production and storage of each beverage. I think that the methodological approach completely allowed this to be done and authors have not fully taken advantage of their own experimental design. Taking this into consideration, stronger outcomes could have been obtained with no additional effort and discussion would have been richer.
AU:
As we explained before, the milk came from only one breed of sheep and it was Polish dairy sheep line 05 with a genetic share of 3/16 Polish Merino and 13/16 East Friesian. The word "genetic share" has been added to this paragraph to make it more communicative.
DISCUSSION
The first paragraphs of this section repeat concepts that have already been introduced and described in detail in the introduction, making some parts parts of the discussion section highly redundant. Please consider rewriting or removing some of these sentences either here or in the introduction section.
AU:
Duplicate section paragraphs have been removed.
CONCLUSIONS
L422-429: This paragraph only states results rather than conclusions and should not be included here. The Conclusions section needs to provide insights about the outcomes of the study and not just provide again the same data from the results section.
AU:
The conclusion supported by the results has been improved. We have removed paragraphs L422-428 from the manuscript. We re-edited the chapter “Conclusion”, removing the typical statements and leaving the conclusions resulting from the realization of the goal of the experiment.
The English language and style:
AU:
The manuscript has been reviewed and edited for the English language by a native speaker, a copyeditor experienced in preparing academic works for the press and for journals. Attached to the cover letter is the Certificate of English Editing.

Reviewer 2 Report
The article deals with an extremely important nutritional problem related to the lactose content in fermented products.
After reading the content, I formulated a few comments regarding the substantive and editorial content. Here they are:
1. I suggest changing the wording of the title a bit (now "Effects of the storage period of sheep milk fermented beverages on lactose content and selected quality parameters"). After all, the authors examined the changes in the lacose content in the products and I believe that it was not related to the influence of storage time - for example, different primers were used for these products and they were different products. The current title indicates that only the storage time was the most important factor influencing the lactose content.
2. Is the calibration method of the Minolta colorimeter described in the methodology correct (line 163): "This was calibrated to following parameters: observer 20, illuminant C"? Please check this record.
3. General remark: please do not begin your descriptions with the wording "Table no ... shows" (lines no. 197 and 222). This is incorrect wording.
4. Note to all tables in the text: please correctly indicate in each caption to the table what the lowercase letters refer to and indicate what the capital letters refer to in the statistical interpretation and which correspond to the results in the row and which to those in the column. It is currently unreadable to the reader.
5. On line 233, which wording is correct: "color" or "colour"? I would even consider changing to "color component parameters" commonly used in professional nomenclature.
6. In Table 3 and Figure 2 the word "coulor" appears - it is definitely not correct. In table 3, the words "consistency" and "smell" are underlined - why?
7. In "Conclusions" the authors refer to the starter preparations used in the production of fermented products (lines 432-435). I believe that this should be dispensed with as there were no others used for each product and there is no comparison in this regard. It was also not one of the goals to be investigated.
8. Literature - the comment concerns the items in the national language of the Authors. Some titles of original publications were translated and stated that they were published in Polish (e.g. 14, 26, 31, 45 and 46). On the other hand, items 16, 17 and 23 were not presented in this way. I am asking for the unification of these provisions.
Author Response
REPLY TO REVIEWER 2
Author’s Notes:
Thank you very much for your useful comments to improve the final version of this manuscript. The manuscript has been modified. Our changes in the text are marked in red.
We have improved the chapter "Introduction" by supplementing it with information on the economic importance of sheep's milk.
We have improved the chapter "Materials and methods" by specifying the description of animal material (only one sheep breed was used), supplementing the information on how to disinfect teats, specifying the chemical analysis method used and adding quotations from professional literature guidelines for the measurement of color using the CIELab color space.
In addition, for better readability, we have included an additional table showing the chemical composition of bulk tank milk (Table 1).
We re-edited the chapter “Conclusion”, removing the typical statements and leaving the conclusions resulting from the realization of the goal of the experiment.
The manuscript has been reviewed and edited for the English language by a native speaker, a copyeditor experienced in preparing academic works for the press and for journals. Attached to the cover letter is the Certificate of English Editing.
on behalf of all authors
Prof. Dr. Jacek Antoni Wójtowski
ORCID: https://orcid.org/0000-0002-9186-006X
Comments and Suggestions for Authors
The article deals with an extremely important nutritional problem related to the lactose content in fermented products.
After reading the content, I formulated a few comments regarding the substantive and editorial content. Here they are:
- I suggest changing the wording of the title a bit (now "Effects of the storage period of sheep milk fermented beverages on lactose content and selected quality parameters"). After all, the authors examined the changes in the lactose content in the products and I believe that it was not related to the influence of storage time - for example, different primers were used for these products and they were different products. The current title indicates that only the storage time was the most important factor influencing the lactose content.
AU:
We followed the reviewer's suggestions by correcting the title of the study to: “Lactose content and selected quality parameters of sheep milk fermented beverages during storage”
- Is the calibration method of the Minolta colorimeter described in the methodology correct (line 163): "This was calibrated to following parameters: observer 20, illuminant C"? Please check this record.
AU:
Sorry, a technical error has occurred - it should be “observer 2o”
General remark: please do not begin your descriptions with the wording "Table no ... shows" (lines no. 197 and 222). This is incorrect wording.
AU:
In our opinion, this is not incorrect wording. The manuscript has been reviewed and edited for English language by a native speaker, a copyeditor experienced in preparing academic works for the press and for journals. Attached to the cover letter is the Certificate of English Editing.
Note to all tables in the text: please correctly indicate in each caption to the table what the lowercase letters refer to and indicate what the capital letters refer to in the statistical interpretation and which correspond to the results in the row and which to those in the column. It is currently unreadable to the reader.
AU:
We are surprised by this remark, especially since this is how we mark the significance of statistical differences by publishing in scientific journals, among others by ELSEVIER publishing houses (Journal of Dairy Science, Measurement and many others).
Nevertheless, we have modified this paragraph. Now it looks like this:
a-d Values in rows with different lowercase letters are significantly different (p<0.05).
A-D Values in columns with different uppercase letters are significantly different (p<0.05).
- On line 233, which wording is correct: "color" or "colour"? I would even consider changing to "color component parameters" commonly used in professional nomenclature.
AU:
The correct spelling of “color” is American English spelling. This spelling is most often used in professional scientific nomenclature and there is no need to change it.
In Table 3 and Figure 2 the word "coulor" appears - it is definitely not correct. In table 3, the words "consistency" and "smell" are underlined - why?
AU:
We apologize. This is a technical error that occurred during the transformation of the text during transmission. We have corrected the error in figure 1 and the underlines in table 3.
- In "Conclusions" the authors refer to the starter preparations used in the production of fermented products (lines 432-435). I believe that this should be dispensed with as there were no others used for each product and there is no comparison in this regard. It was also not one of the goals to be investigated.
AU:
We believe that the assessment of the quality parameters fully allows us to conclude that the starter cultures YO 122 and ABY (Biochem), as well as LYOFAST Y 480F and LYOFAST MT 036LX (Sacco), were fully suitable for the production of fermented beverages from sheep's milk. This is evidenced by the very good quality parameters of finished products obtained in the research. With such a conclusion, no additional comparison is needed in this respect. For thirty years, the corresponding author of the article has been dealing with the issues of obtaining and processing sheep's milk. It is i.a. co-author of recipes for products from this type of milk. Based on many years of experience, I conclude that rarely commercial starter cultures intended for the inoculation of cow's milk are fully suitable for the production of sheep's milk products.
As for the reviewer's statement: It was also not one of the goals to be investigated - quite the opposite. In the paragraph on the aim of the present study (L. 100-102) we include the following wording: “An equally important aspect of the research was to assess the suitability of starter cultures intended for the production of fermented beverages from cow’s milk in the production of sheep’s milk products”.
- Literature - the comment concerns the items in the national language of the Authors. Some titles of original publications were translated and stated that they were published in Polish (e.g. 14, 26, 31, 45 and 46). On the other hand, items 16, 17 and 23 were not presented in this way. I am asking for the unification of these provisions.
AU:
As suggested by the reviewer, the spelling of the literature for items 16, 17 and 23 was unified.
The English language and style:
AU:
The manuscript has been reviewed and edited for English language by a native speaker, a copyeditor experienced in preparing academic works for the press and for journals. Attached to the cover letter is the Certificate of English Editing.

Reviewer 3 Report
1. What is the average weight of ewes?
2. Is the feeding diet for all ewes the same?
3. Please write the chemical analysis of the used milk in a table.
4. It is better to write the scientific names of bacteria in italics.
5. Why is there no complete chemical analysis for the different types of yogurt at the beginning of the experiment (the first week) and at the end of the experiment (week 21)? To clarify the total differences between the types of yogurt
6. In line 318, change (In our research) to (in this study).
7. In lines 331,430, do not use the personal pronouns (our).
Author Response
REPLY TO REVIEWER 3
AU: We wish to thank you very much for the very thorough and constructive review, detailed, critical comments and suggestions, which proved very helpful in the preparation of the corrected version of the study.
The manuscript has been reviewed and edited for the English language by a native speaker of English and a copyeditor experienced in preparing academic works for the press and for journals. Attached is the Certificate of English Editing.
on behalf of all authors
Prof. Dr. Jacek Antoni Wójtowski
ORCID: https://orcid.org/0000-0002-9186-006X
Comments and Suggestions for Authors
What is the average weight of ewes?
AU:
The ewes body weight was 61.0 ± 2,5 kg (L. 110)
Is the feeding diet for all ewes the same?
AU:
Yes, the feeding diet for all ewes was the same.
Please write the chemical analysis of the used milk in a table.
AU:
Was done. Modified as requested.
It is better to write the scientific names of bacteria in italics.
AU:
Sorry, a technical error has occurred. The spelling was corrected.
Why is there no complete chemical analysis for the different types of yogurt at the beginning of the experiment (the first week) and at the end of the experiment (week 21)? To clarify the total differences between the types of yogurt
AU:
The aim of the research was only to evaluate changes in lactose concentration and selected physicochemical, rheological and organoleptic parameters of fermented milk beverages. When performing such an assessment, a comprehensive chemical analysis of the products is not performed.
In line 318, change (In our research) to (in this study).
AU:
Was done. Modified as requested.
In lines 331,430, do not use the personal pronouns (our).
W wierszach 331.430 nie używaj zaimków osobowych (nasz).
AU:
Was done. Modified as requested.
The English language and style:
AU:
The manuscript has been reviewed and edited for English language by a native speaker, a copyeditor experienced in preparing academic works for the press and for journals. Attached to the cover letter is the Certificate of English Editing.

Round 2
Reviewer 1 Report
Dear authors,
After revising this new version of the manuscript Animals-1962295, entitled “Effects of the storage period of sheep milk fermented beverages on lactose content and selected quality parameters" and your replies, I reckon all my queries and questions have been adequately addressed. Thus, I have no further comments.
All the very best.